# Identification of a New *Pathogenic fungi* Causing Sorghum Leaf Spot Disease and Its Management Using Natural Product and Microorganisms

**DOI:** 10.3390/microorganisms11061431

**Published:** 2023-05-29

**Authors:** Guoyu Wei, Wei Zhao, Anlong Hu, Mingjian Ren, Yunxiao Huang, Huayang Xu

**Affiliations:** College of Agriculture, Guizhou University, Guiyang 550025, China; wgy18722856021@163.com (G.W.); zw18786470874@163.com (W.Z.); rmj72@163.com (M.R.); hyxluck@163.com (Y.H.); xuhuayang0815@163.com (H.X.)

**Keywords:** antagonistic bacteria, natural product, leaf spot, sorghum, *F. thapsinum*

## Abstract

*Sorghum bicolor* is cultivated worldwide. Leaf spot of sorghum, which leads to leaf lesions and yield reduction, is a prevalent and serious disease in Guizhou Province, southwest China. In August 2021, new leaf spot symptoms were observed on sorghum leaves. In this study, traditional methods and modern molecular biology techniques were used to isolate and identify the pathogen. Sorghum inoculated with the isolate GY1021 resulted in reddish brown lesion that similar to symptoms observed in the field: the original isolate inoculated was reisolated and Koch’s postulates were fulfilled. Based on morphological features and phylogenetic analysis of the internal transcribed spacer (ITS) combined sequence with β-tubulin (*TUB2*) and translation elongation factor 1-α (*TEF-1α*) genes, the isolate was identified as *Fusarium thapsinum* (Strain accession: GY 1021; GenBank Accession: ITS (ON882046), *TEF*-1α (OP096445), and β-*TUB* (OP096446)). Then, we studied the bioactivity of various natural products and microorganisms against *F. thapsinum* using the dual culture experiment. Carvacrol, 2-allylphenol, honokiol, and cinnamaldehyde showed excellent antifungal activity, with EC_50_ values of 24.19, 7.18, 46.18, and 52.81 µg/mL, respectively. The bioactivity of six antagonistic bacteria was measured using a dual culture experiment and the mycelial growth rate method. *Paenibacillus polymyxa*, *Bacillus amyloliquefaciens* and *Bacillus velezensis* displayed significant antifungal effects against *F. thapsinum*. This study provides a theoretical basis for the green control of leaf spot of sorghum.

## 1. Introduction

*Sorghum bicolor* is the fifth-largest crop in the world with high nutritional value. Their seeds contain roughly 70% starch and 11% protein, and are rich in vitamins, minerals, and micronutrients [1]. Sorghum has a wide range of applications, including brewing, food processing, and feed. Sorghum has strong resistance to stress, such as drought, cold, salt and alkali [2]; furthermore, it is adapted to a wide range of soil types and pH. However, unscientific use of fungicides, large area monoculture, climatic changes, and other related factors induced the spread of diseases, which have great influence on sorghum yield. Leaf spot is one of the most damaging diseases. The disease is mainly to harm the leaves. The leaves at the beginning of the onset have irregular spots, and then the spots are expanded to induce severe necrosis, which reduces the yield and quality of the sorghum [3]. Leaf spot usually leads to be 12–55% economic loss of sorghum production [4].

There are many fungi which were reported be the pathogen of sorghum leaf spot disease, for example, *Pestalotiopsis trachycarpicola* [1], *Alternaria* spp. [5], and *Curvularia* spp. In 2018, planting area of sorghum was more than 600,000 ha with a production of approximately 290 million tons in China. At present, sorghum leaf spot disease has occurred frequently in major sorghum producing areas, and has tended to be worse each year in China, especially in Guizhou province. Carbendazim, tectol and benendazim are the most commonly used chemical fungicide to control leaf spot of sorghum [6]. However, the use of chemical fungicides not scientific and rational result residue, resistance, environmental pollution and other problems [7], Halko et al. indicated that benzimidazole fungicides residue in soil, Gabriel et al. determined residue levels of seven commonly used pesticides and suggested long-term exposure can still cause health risks. Lim et al. studied and reported that the benomyl metabolite carbendazim has reproductive toxicity, Younghae et al. found that if applied unscientifically, which can lead to fungicide resistance in *F. fujikuroi* [8,9,10]. In contrast, natural products, such as natural product and antagonistic microbes, have the advantages of low toxicity, cause no pollution, and do not easily lead to pesticide resistance.

Compared with chemical control, biological control is considered a safe and effective method in plant diseases controlling. In recent years, many scholars have reported that pathogens can be controlled by biological agents. These include *Bacillus amyloliquefaciens,* which could control *Pectobacterium carotovorum* in potatoes [11]: Yuan et al. found the biocontrol mechanism of *Paenibacillus polymyxa* [12], Ali et al. revealed the high potential of biocontrol in *Paenibacillus polymyxa* complex [13], Zhou et al. reported *Bacillus velezensis* which could control the tomato bacterial wilt disease [14], and *Bacillus licheniformis* PR2 which could control fungal diseases and increase production [15]. Nonetheless, there has been no research into the biological control of sorghum leaf spot. Meanwhile, as natural products are also safe and effective for controlling plant disease, many studies have examined natural products as potential substances for the control of plant diseases and a good number of natural products can control plant diseases. Liu et al. reported that carvacrol could control vegetable diseases [16]. Qu et al. studied and found that 2-allylphenol showed excellent bioactivity against several plant pathogens [17], Oufensou et al. reported that honokiol had a strong antifungal effect against *Fusarium* [18]. It is reported that cinnamaldehyde can inhibit *F. sambucinum* ergosterol biosynthesis [19]. Mo et al. indicated that natural products magnolol is active against *Rhizoctonia solani*, and its mechanism lies in the destruction of the plasma membrane [20].

Liquor making is one of the oldest industries in the world [21]. At present, organic sorghum is used as the main raw material for liquor making in China. In addition, with the development of the national economy, the demand for organic sorghum has increased. Therefore, high-quality organic sorghum is urgently needed by brewing enterprises, and there is a need to develop efficient, safe, green, and sustainable prevention and control measures. Given the above situation, the causative pathogen of sorghum leaf spot disease was isolated and identified, then we screened biocontrol bacteria (*Paenibacillus polymyxa*, *Bacillus velezensis*, *Bacillus amyloliquefaciens*, *Bacillus subtilis*, *Bacillus megaterium*, and *Bacillus licheniformis*) and natural products with the potential to prevent and control sorghum leaf spot disease. In this study, our aim was to provide a “green” management method against sorghum leaf spot disease. Therefore, we explored the pathogens of sorghum leaf spot disease and screened the potential *Bacillus* spp. and natural fungicidal agents. This study offers new biological control resources against *Fusarium thapsinum*.

## 2. Materials and Methods

### 2.1. Pathogens, Antagonistic Strains, and Natural Products

The leaves of sorghum were collected (Table 1) and soaked in 75% alcohol for 30 s, disinfected with 10% sodium hypochlorite for 1–2 min, cleaned with sterile water thrice, drained with sterile paper, and dried in sterile Petri dishes. The dried symptomatic tissues were then transferred to potato dextrose agar (PDA: 200.0 g potato, 20.0 g glucose, 17.0 g agar/L) plates on an ultra-clean workbench and cultured in a biochemical incubator (Ningbo Jiangnan Instrument Co., Ltd., Zhejiang, China) at 25 °C. After incubation at 25 °C for 48 h, a single colony was transferred to new plates, incubated at 25 °C for 5 days. The single colony was soaked in 30% glycerol and stored at −80 °C. Then, according to Koch’s postulates, isolates were inoculated on sorghum leaves, and re-isolated and purified. *Paenibacillus polymyxa* Y-1, *Bacillus velezensis* MT310, *Bacillus amyloliquefaciens* MT323, *Bacillus subtilis* MT332, *Bacillus megatherium* L2, *Bacillus licheniformis* MTQ23 all are from the College of Agriculture, Guizhou University. Natural products (eugenol, magnolol, thymol, cinnamaldehyde, honokiol, carvacrol, 2-allylphenol) with purities of ≥98% were purchased from Aladdin Reagent Co., Ltd. (Shanghai, China) and stored at 2–8 °C.

### 2.2. Pathogenicity Assays

Five fungal strains were isolated for Koch’s hypothesis [22]. The fungal isolates were incubated in potato dextrose broth (PDB; 200.0 g potatoes infusion, 20.0 g glucose, 1 L water), in 120 rpm and 25 °C shakers for 5 d, then, conidia were collected by filtering through gauze. Then, a concentration of 1 × 10^6^ conidia/mL was collected with a hemocytometer and spray 500 μL evenly with a sterile watering can on healthy sorghum leaves. 500 μL of sterile distilled water was sprayed as a blank control. The inoculated sorghum was placed in an incubator under 28 °C, 85% RH and 12/12 h light/dark, and occurrence of disease was regularly observed with phytopathometric assessments. Experiments were performed in triplicate.

### 2.3. Pathogen Identification

The identification of pathogens usually includes both morphology and molecular biology. The morphology of the mycelia and fungal spores incubated at 28 °C for 10 days was observed using an optical microscope (LEICA ICC50 W, Leica Microsystems Co., Ltd., Shanghai, China). Morphological identification of the isolates was performed [23]. Fungal DNA was extracted from fresh aerial mycelia grown on PDA plates using a Fungal Genomic DNA Kit (Tiangen Biotech (Beijing, China) following the manufacturer’s instructions. Universal primers targeting ITS, *TEF-1α, and β-TUB* genes (Table 2) were used in polymerase chain reaction (PCR) program following Watanabe’s method [24]. DNA amplification was performed in a final volume of 25 μL containing 12.5 μL of 2×Taq Master Mix (Sangong Bioengineering Co., Ltd. (Shanghai, China), 10 μM of each forward and reverse primer, 100 ng of DNA template, and ddH_2_O. The amplified PCR products were sequenced by Sangon Biotech Co., Ltd. (Shanghai, China). Then a polygene phylogenetic tree was constructed using the maximum likelihood (ML) method in MEGA 7.0 [25] software with bootstrap values based on 1000 replications. *F. delphinoides* CBS 110140 was used as an outgroup.

### 2.4. Antagonistic Bacteria Screening

The antagonistic bacteria were tested in vitro for the inhibition effect against *F. thapsinum* [27]. A 5 mm disc from the edge of colony was transferred to the center of the PDA plate. Then, 0.5 cm circles were punched into the filter paper and soaked in the antagonistic strains. The antagonistic strains were inoculated at four symmetrical points 20 mm away from the *F. thapsinum*. The control inoculated 0.5 cm filter paper circles with sterile distilled water at four symmetrical points located 20 mm from the pathogen. All plates were incubated for 5 days under 28 °C. The mycelial growth inhibition rate was calculated according to formula: A = (m_1_ − m_2_)/m_1_ × 100, where A is mycelia growth inhibition rate, m_1_ is the hyphae area of the control, m_2_ is the hyphae area of the treated [28]. All experiments were repeated three times.

### 2.5. Antifungal Activity of Aseptic Filtrate from Antagonistic Bacteria against GY 1021

The inhibition effect of aseptic filtrate on the mycelial growth of *F. thapsinum* was assessed according [29]. An amount of 1 mL of antagonistic bacteria with OD (optical density) = 0.6 was transferred to a 0.25 L flask with 0.1 L nutrient agar broth (NA). The conical flasks sealed with parafilm were incubated in a rotary shaker (150× *g* at 37 °C). After 48 h of incubation, bacterial solution was filtered using a sterile filter gauze, followed by filtration through a 0.44 µm and a 0.22 µm membrane filter, then, aseptic filtrate was obtained. In addition, the sterile filtrate was added to the medium to concentrations of 25 mL L^−1^, 50 mL L^−1^, and 100 mL L^−1^, respectively, the medium was added to plates, after the medium became cool and solidify, the pathogen discs were inoculated to the center of PDA plates containing. After incubation under 25 °C for 7 days, the colony diameter was measured using the “ten” crossing method. All experiments repeated three times. Adding sterile water was used as a control. The inhibitory rate was calculated according: inhibitory rate (%) = [(dcontrol − dtreatment)/dcontrol] × 100 [30], d is the diameter of the *F. thapsinum* colony.

### 2.6. Antimicrobial Activity of Natural Products on Mycelial Growth

The antifungal activity of natural products was determined by mycelium growth rate method [31], different natural products were dissolved in appropriate organic solvents (2-allylphenol and eugenol in ethanol, honokiol in dimethyl sulfoxide, cinnamaldehyde, carvacrol, and magnolol in acetone, thymol in sterile water, all solvents have a concentration greater than 99%) and then, using water, mixed evenly with the PDA medium at different concentrations. The control contains the corresponding solvent. After incubation for 7 days at 25 °C, the colony diameter was measured using the "ten" crossing method. The EC_50_ (concentration for 50% of maximal effect) values of different natural antifungals were calculated using IBM SPSS analytics (SPSS Inc., Chicago, IL, USA) [20]. Each treatment was conducted with three replicates.

### 2.7. Effects of Aseptic Filtrate and Natural Products on Controlling Sorghum Leaf Spot Disease

Four holes were punctured using a sterile syringe at suitable locations in the sorghum leaves. Then, different antagonistic bacteria of aseptic filtrate (100 mL L^−1^) were sprayed on each leaf, and three natural products (2-allylphenol, carvacrol, honokiol) were formulated at 7, 24, and 46 mg/L, respectively, and the liquid medicine was sprayed evenly on each leaf. An amount of 1 × 10^6^ conidia/mL was inoculated at the site of the stab wound. Subsequently, the plants were placed in a sterile crisper, placed into greenhouses, and maintained under a fixed photoperiod (12 h light, 12 h dark) at 28 °C and 80% relative humidity. After incubation for 7 days, the lesion area was measured using ImageJ software (National Institute of Health, Bethesda, MD, USA). The disease incidence (%) was calculated using the formula: (M_1_ − M_2_)/M_1_ × 100, where M_1_ and M_2_ represent the decayed areas in the treatment group and control group, respectively. Experiments were performed in triplicate.

### 2.8. Statistical Analyses

All statistical data analyses were performed using Excel 2021. One-way ANOVA was performed as per Duncan’s multiple range test to determine the significance of differences (*p* < 0.05 were considered significant). Charts were plotted with Origin 2021 and DPS.

## 3. Results

### 3.1. Isolation of the Pathogen and Pathogenicity Assays

The diseased leaves were found in a sorghum plantation in Huaxi County, the diseased leaves showed red necrotic spots and irregular lesions on the leaves (Figure 1a). Five isolates were isolated and purified. The five isolates were inoculated on sorghum leaves: only the strain GY1021 caused spots, which was similar with the symptoms in the field, and the same fungi were reisolated. Pathogenicity results showed that 6 d after treatment with the strain GY1021 conidia suspension, red irregular spots appeared on the sorghum leaves and the leaf margins withered and shrunk (Figure 1b,c), after 10 days, the lesion became gray and the whole leaf withered (Figure 1c). Figure 1d is control. The leaves inoculated with sterile distilled water did not show leaf spot symptoms. The same strains were isolated from diseased leaves and satisfied Koch’s hypothesis.

### 3.2. Morphological Identification and Molecular Identification of the Pathogen

According to the morphological characteristics of colony, the pathogen was identified as *Fusarium* spp. The characteristics of the colony of *Fusarium* species are shown in Figure 2, and the shape land septum of macroconidia are also described as shown. The *Fusarium* colonies grew quickly (0.76 mm d^−1^ at 28 °C) and were usually white or yellow in appearance with developed aerial mycelia. The new colonies cultured on PDA were white, whereas the old colonies cultured for 5 days were yellow in the middle with a white fringe (Figure 2A,B). Conidia were produced on water agar containing carnation leaf pieces, and the conidiogenous cells were phialidic. Under optical microscopy, the conidia appeared slightly curved, with blunt and occasionally hooked apical cells. There were many small conidia, and the large, mature conidia were usually three-septate and measured 29.32–43.89 µm × 3.63–5.52 µm.

For phylogenic analysis, the phylogenetic tree was constructed by using the combined sequences of ITS, *β-TUB* and *TEF-1α* in MEGA 7.0 software with bootstrap values based on 1000 replications, as shown in Figure 3, including 1 species of *Fusarium thapsinum*, 18 other referred isolates of *Fusarium* species, and *F. delphinoides* was an outgroup species (Table 3). The sequences of representative isolates of ITS, *TUB2*, and *TEF-1α* were aligned with the corresponding sequences of different species of *Fusarium* obtained from NCBI’s GenBank nucleotide database.

To identify the strain GY 1021, the sequences of the ITS (557), *tef* (694), and *β-tub* (974) were amplified by PCR and then submitted to GenBank database with the accession numbers of ITS (ON882046), *tef-1α* (OP096445), and *β-tub* (OP096446), respectively. The consequences of BLAST displayed that the ITS sequence of the strain GY 1021 had 100% identity with *Fusarium thapsinum* (KR071692), the *tef-1α* sequence of strain GY 1021 had 100% identity with *Fusarium thapsinum* (MW401962), and the *β-tub* sequence of strain GY 1021 had 100% identity with *Fusarium thapsinum* (MW402222). The maximum likelihood (ML) method in MEGA 7.0 software was used to construct a multigene phylogenetic tree. Phylogenetic analysis further demonstrated that strain GY 1021 was clustered with *Fusarium thapsinum* (Figure 3). Through detailed morphological studies and phylogenetic tree construction, strain GY1021 was finally identified as *Fusarium thapsinum*. Note: GY 1021 is the pathogen (*F. thapsinum*). *F. delphinoides*(CBS 110140) is outgroup.

### 3.3. Screening of Antagonistic Strains

The results of dual culture experiment displayed that *Paenibacillus polymyxa* have strong inhibitory activity against *F. thapsinum.* The best inhibitory effect was obtained with *Paenibacillus polymyxa*, and the inhibitory effect on mycelial growth was approximately 82.12% (Figure 4), the inhibitory rates of *Bacillus amyloliquefaciens*, *Bacillus velezensis*, *Bacillus subtilis*, *Bacillus megaterium*, and *Bacillus licheniformis* was 77.05%, 69.70%, 61.07%, 61.67%, and 64.56%, respectively.

### 3.4. Effects of Aseptic Filtrate on the Mycelial Growth of F. thapsinum

The inhibitory effect of the aseptic filtrate on *F. thapsinum* were detected in vitro (Table 4). The results indicated that the antibacterial activity increased with the increase in the concentration of sterile filtrate. The mycelial growth of *F. thapsinum* was significantly suppressed by aseptic filtrate of *Bacillus amyloliquefaciens, Paenibacillus polymyxa*, *Bacillus velezensis, Bacillus subtilis, Bacillus licheniformis*, and *Bacillus megaterium*, with inhibition rates of 61.70% ± 0.40%, 57.91% ± 0.40%, 49.87% ± 0.23%, 44.44% ± 0.23%, 43.49% ± 0.77%, and 41.60% ± 0.23%, respectively. The aseptic filtrate of *Bacillus amyloliquefaciens* resulted in the greatest inhibition of *F. thapsinum.*

### 3.5. Antifungal Activity of Natural Products on Mycelium

The screening of antifungal activity in natural products against *F. thapsinum* strain is conducive to the development of green, low-toxicity fungicides. The results of the sensitivity testing of seven natural products agents on *F. thapsinum* are shown in Table 5. 2-allylphenol had significant antifungal activity, with an EC_50_ value of 7.18 ± 0.02 µg/mL, followed by carvacrol, honokiol, cinnamaldehyde, magnolol, and eugenol with EC_50_ values of 24.19 ± 0.49, 46.18 ± 0.60, 52.81 ± 0.30, 75.78 ± 0.22, and 82.39 ± 0.17 µg/mL, respectively. Thymol antibacterial activity was poor, with an EC_50_ value of 92.84 ± 0.51 µg/mL. When examining the regression equations, the slope of 2-allylphenol was the largest, at 2.4329, and that of thymol was the smallest, at 1.3692. This indicated that *F. thapsinum* was most sensitive to 2-allylphenol and least sensitive to thymol.

### 3.6. Effects of Aseptic Filtrate and Natural Products on Controlling Sorghum Leaf Spot Disease

To confirm the effect of the aseptic filtrate and natural products on *F. thapsinum* in vivo, the aseptic filtrate of *Bacillus amyloliquefaciens*, *Bacillus velezensis*, *Paeni Bacillus polymyxa*, and the natural products 2-allylphenol, carvacrol, and honokiol were used to alleviate the severity of sorghum decay caused by *F. thapsinum* (Figure 5). AF (aseptic filtrate); NA (natural antifungal agent). *Bacillus amyloliquefacien, Bacillus velezensis*, and *Paenibacillus polymyxa* are all at 100 mL L^−1^, 2-allylphenol, carvacrol, and honokiol are 7, 24, and 46 mg L^−1^, respectively. Disease inhibition rate was measured after treatment with three antagonistic bacteria aseptic filtrates and three natural products. After 15 days, the spots were measured. The disease inhibition rates were 78.97%, 55.62%, and 69.51% for *Bacillus amyloliquefaciens*, *Bacillus velezensis*, and *Paenibacillus polymyxa*, respectively, and the disease inhibition rates were 87.29%, 85.23%, and 80.13% for 2-allylphenol, carvacrol, and honokiol, respectively. *Bacillus amyloliquefaciens, Paenibacillus polymyxa,* 2-allylphenol, carvacrol, and honokiol have significant inhibitory effects (Figure 6). The lesion area was measured using ImageJ software (National Institute of Health, Bethesda, MD, USA).

## 4. Discussion

Sorghum planting has a long history and a wide range of regions. It is a very important raw material for grain, feed crops, as well as winemaking and vinegarmaking. However, sorghum is susceptible to various pathogens during the processes of cultivation, transportation, and storage, resulting in a serious decline in the quality of sorghum and enormous economic losses. Diseases reported so far include anthrax [32], leaf spot, and stem rot; stem rot was caused by *F. thapsinum*, *Pestalotiopsis trachycarpicola* [3]. *Fusarium thapsinum* is a common and highly pathogenic pathogen. At present, the application of chemical fungicides is still the main measure to control sorghum diseases. However, long-term application of chemical pesticides will endanger food safety and destroy environment. Accordingly, it is particularly necessary to search green and effective measures to control sorghum diseases. Biological control can promote crop growth, increase beneficial microbial populations, and effectively control crop diseases [31]. Therefore, the study of natural products and biological means to control sorghum diseases is of great significance. In general, natural products control the occurrence and prevalence of diseases by inhibiting the growth and reproduction of pathogens. Wei et al. found that cinnamaldehyde first significantly affects the integrity of the cell membrane of Fusarium sambucinum, and then reduces the mitochondrial membrane potential and induces the accumulation of reactive oxygen species in cells [19]. At the same time, cinnamaldehyde can reduce ergosterol content. The results of Mo et al. showed that honokiol can significantly damage the plasma membrane of *R. solani*, and interfere with cell respiratory metabolism, thus inhibiting the growth of mycelium [20]. Microbial antagonists control disease by competing with pathogens for nutrients and space; nonpathogenic *Fusarium cereus* competes with pathogenic *Fusarium dysphoria* to inhibit blight by competing for nutrients such as carbon and iron [33]. Kalantarl et al. confirmed that a mixture of *Bacillus, Pseudomonas*, and *Rhizobia* had a synergistic effect, producing a variety of beneficial substances that could effectively inhibit blight root rot and increase soybean yield [34]. The research data of Azaiez et al. showed that protection against potato soft rot disease may be related to glycolipid production by Bacillus amyloliquefaciens [11]. Bacillus velezensis can both individually promote tomato plant growth, increase leaf chlorophyll content, enhance defensive enzyme activities, and induce the accumulation of D-fructose and D-glucose [14].

In this study, a pathogen of sorghum leaf disease was isolated, *F. thapsinum*, from sorghum “hongyingzi”. Currently, there are almost no report of the effect of *F. thapsinum* on sorghum; however, its impact cannot be ignored, and prevention measures should be taken in time. Presently, chemical control is the main measure to prevent and treat diseases caused by *F. thapsinum*. Chemical fungicides agents could effectively control plants diseases; however, unreasonable application of chemical fungicides can result in pathogens resistance [35]. In contrast, biocontrol bacteria agents and natural products have attracted more and more attention because they are in line with the European IPM framework and organic agriculture legislation while improving the sustainability of agricultural operations [36]. Therefore, we screened natural products and biocontrol bacteria. The study results show that these natural products and biocontrol bacteria could effectively inhibit the growth of *F. thapsinum*. However, the effect of natural products was notably better than that of aseptic filtrates. There are several possible explanations for this: (1) There are defects in the preparation method of aseptic filtrates that should be improved. (2) The aseptic filtrate has a low content of active ingredients and may need to be concentrated. (3) The antibacterial components of the aseptic filtrate may be volatilized or decomposed. Natural products and biocontrol bacteria have significant inhibitory effects on *F. thapsinum*; however, their success in controlling the sorghum disease caused by *F. thapsinum* still needs to be verified by field experiments. The antibacterial mechanism of these natural products and some biocontrol bacteria is not clear; thus, further study is required.

## 5. Conclusions

In this study, a new pathogen GY 1021 causing leaf spot of sorghum was isolated from sorghum “Hongyingzi,” and was identified as *F. thapsinum*. Simultaneously, we screened the inhibitory effects of seven natural products on *F. thapsinum*, 2-allylphenol, carvacrol, and honokiol were found to have the potential to inhibit the growth of mycelium, in vitro, *F. thapsinum* was most sensitive to 2-allylphenol. In the screening of six biocontrol bacteria, *Paenibacillus polymyxa* was found to be the most effective antifungal agent against *F. thapsinum*. The aseptic filtrate of *Paenibacillus polymyxa* caused the highly inhibition of *F. thapsinum*. Honokiol, 2-allylphenol, carvacrol and the aseptic filtrate of *Paenibacillus Polymyxa* could effectively inhibit the spots extension. This study has certain reference value for the prevention and control of sorghum leaf diseases.

## Figures and Tables

**Figure 1 microorganisms-11-01431-f001:**
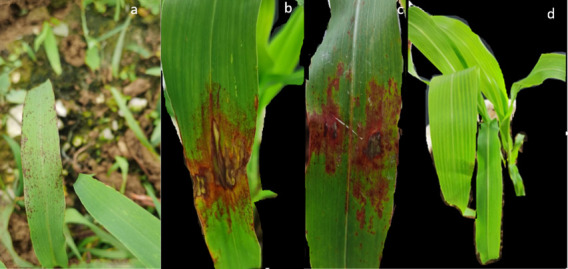
Natural field symptoms of sorghum leaf disease (**a**); Symptoms 6 days after artificial inoculation (**b**); Symptoms 10 days after artificial inoculation (**c**); Control (**d**).

**Figure 2 microorganisms-11-01431-f002:**
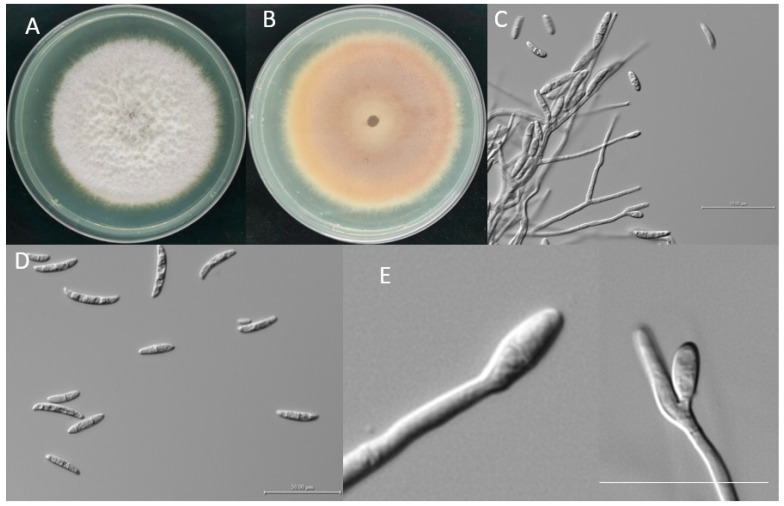
The morphology of pathogen grown on PDA medium for 5 days (**A**,**B**); conidiophores (**D**); the structure that produces conidia (**C**,**E**). Scale bar is 50 μm.

**Figure 3 microorganisms-11-01431-f003:**
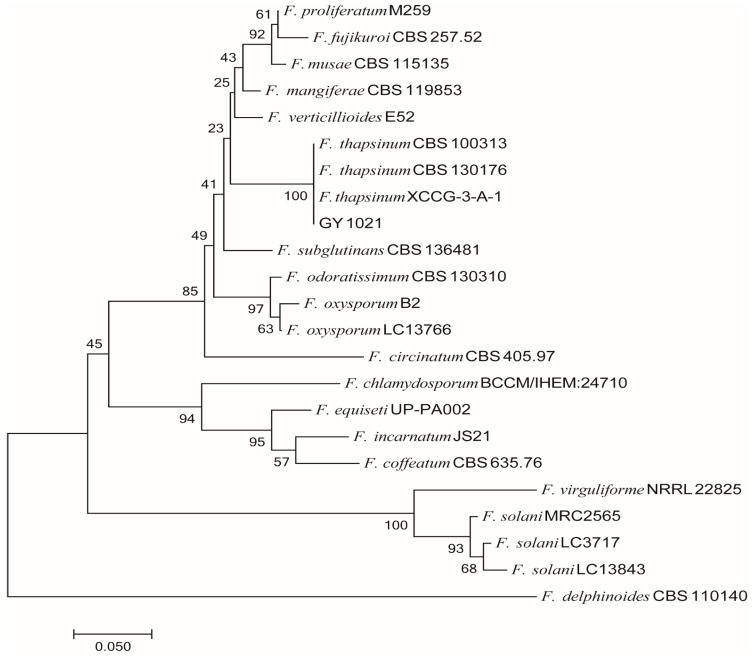
The ITS–*TEF-1α–TUB2* phylogenetic tree.

**Figure 4 microorganisms-11-01431-f004:**
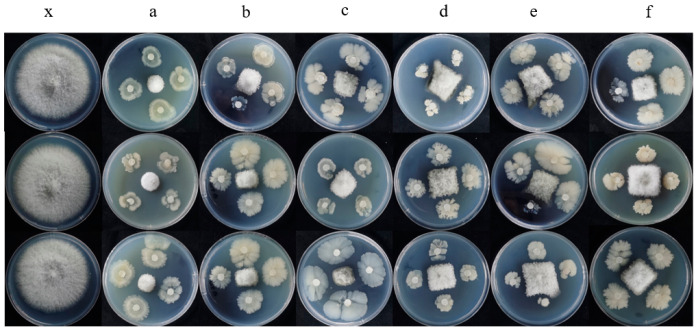
The antagonistic bacteria were tested in vitro for the inhibitory effect against *F. thapsinum*. (**a**) *Paenibacillus polymyxa*; (**b**) *Bacillus velezensis*; (**c**) *Bacillus amyloliquefaciens*; (**d**) *Bacillus subtilis*; (**e**) *Bacillus megaterium*; (**f**) *Bacillus licheniformis*; (**x**) Control.

**Figure 5 microorganisms-11-01431-f005:**
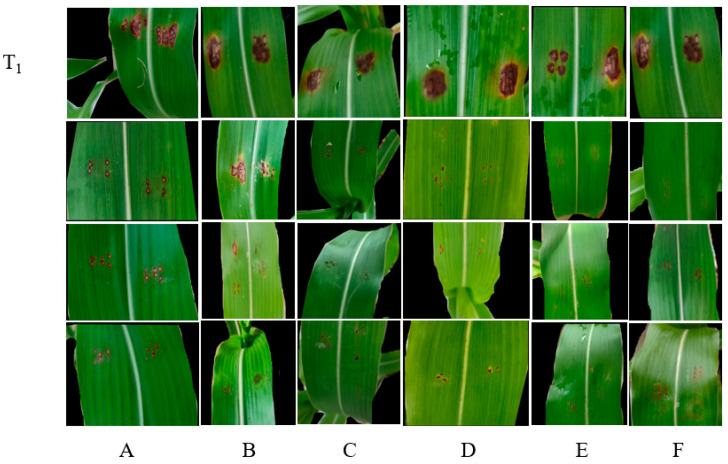
In vivo antifungal effects of aseptic filtrate and natural products on disease extension in sorghum caused by *F. thapsinum*. (**A**) (*Bacillus amyloliquefaciens*), (**B**) (*Bacillus velezensis*), (**C**) (*Paenibacillus polymyxa*), (**D**) (2-allylphenol), (**E**) (carvacrol), (**F**) (honokiol), (**T_1_**) (control).

**Figure 6 microorganisms-11-01431-f006:**
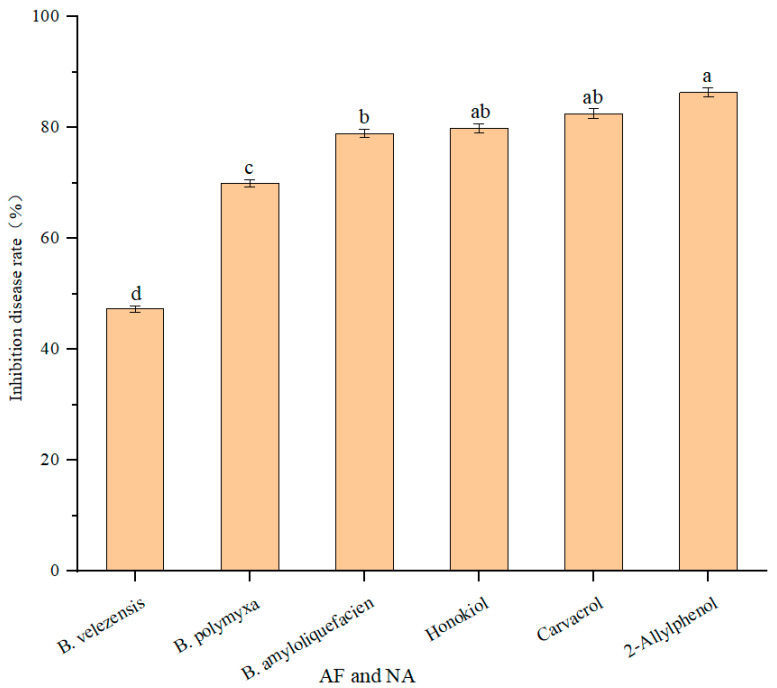
In vivo effect of aseptic filtrate and natural products on disease extension in sorghum leaf spot disease caused by *F. thapsinum*. Data are displayed as the mean ± SE. According to Duncan’s multiple range test. The same lowercase letters represent no significant difference between treatments (*p* < 0.05).

**Table 1 microorganisms-11-01431-t001:** Source of symptomatic leaves.

Place	Time	Longitude and Latitude
Huaxi City, Guizhou Province, China	August 2021	26°4′ N, 106°6′ E

**Table 2 microorganisms-11-01431-t002:** PCR primers for ITS, *TEF-1α*, and *β-TUB* gene amplification.

Target Sequence	Primer	Primer Sequence (5′–3′)
ITS	ITS1	TCCGTAGGTGAACCTGCGG
	ITS4	TCCTCCGCTTATTGATATGC
*TEF* [26]	EF1	ATGGGTAAGGAGGACAAGAC
	EF2	GGAGGTACCAGTGATCATGTT
*TUB2* [26]	Bt2a	GGTAACCAAATCGGTGCTGCTTTC
	Bt2b	ACCCTCAGTGTAGTGACCCTTGGC

**Table 3 microorganisms-11-01431-t003:** Reference isolates used in this study and their GenBank accession numbers.

Species	Strain Accession	GenBank Accession		
		ITS	TEF-1α	β-TUB
*F. thapsinum*	CBS00313	-	MW401962	MW402163
*F. thapsinum*	CBS130176	KR071692	MW402022	MW402222
*F. thapsinum*	XCCG-3-A-1	-	MT997082	-
*F. verticillioides*	E52	KJ467098	KJ555083	KJ544181
*F. oxysporum*	B2	MZ060273	MN754062	MN754078
*F. proliferatum*	M259	KJ467095	KJ555080	KJ544176
*F. musae*	CBS 115135	KR071632	KR071710	-
*F. fujikuroi*	CBS 257.52	MH857023	MW402119	KU603885
*F. subglutinans*	CBS 136481	KR071625	KU711692	KU603893
*F. odoratissimum*	CBS 130310	-	MH485013	MH485104
*F. virguliforme*	NRRL 22825	GU170655	EF408437	EF408472
*F. circinatum*	CBS 405.97	MH862654	KM231943	KM232080
*F. incarnatum*	JS21	MT889974	MT895846	MT895843
*F. coffeatum*	CBS 635.76	MH861016	MN120755	-
*F. chlamydosporum*	BCCM/IHEM:24710	KJ125534	KJ126126	KJ125830
*F. mangiferae*	CBS 119853	MH863065	MN534016	MN534140
*F. equiseti*	UP-PA002	MH521295	MH521297	MH521296
*F. solani*	LC3717	MW016736	MW620197	MW534074
*F. solani*	LC13843	MW016729	MW620190	MW534069
*F. oxysporum*	LC13766	MW024413	MW594353	MW533955
*F. solani*	MRC2565	MH584200	MH582420	
*F. delphinoides*	CBS 110140	EU926235	EU926302	EU926368

**Table 4 microorganisms-11-01431-t004:** In vitro inhibitory activity.

Strain Name	Treatment (mL L^−1^)	Colony Diameter (mm)	Inhibition Rate (%)
*Paeni* *bacillus polymyxa*	100	32.1 ± 0.2 g	61.7 ± 0.4 a
	50	43.0 ± 0.2 f	46.1 ± 0.4 a
	25	48.0 ± 0.2 f	39.0 ± 0.4 b
*Bacillus amyloliquefaciens*	100	34.6 ± 0.3 f	57.9 ± 0.4 b
	50	43.2 ± 0.3 f	45.6± 1.1 b
	25	46.1 ± 0.3 g	41.8 ± 0.7 a
*Bacillus* *velezensis*	100	40.3 ± 0.3 e	49.9 ± 0.2 c
	50	49.1 ± 0.3 e	37.3 ± 0.2 c
	25	62.0 ± 0.2 c	19.1 ± 0.4 e
*Bacillus subtilis*	100	44.1 ± 0.3 d	44.4 ± 0.2 d
	50	50.0 ± 0.3 d	36.2 ± 0.4 d
	25	54.0 ± 0.2 e	29.3 ± 0.5 c
*Bacillus licheniformis*	100	45.2 ± 0.3 c	43.5 ± 0.8 e
	50	60.4 ± 0. 4 b	21.3 ± 0.8 f
	25	71.1 ± 0.3 b	6.1 ± 0.2 f
*Bacillus* *megaterium*	100	46.0 ± 0.3 b	41.6± 0.2 f
	50	55.8 ± 0.3 c	27.9 ± 0.2 e
	25	61.0 ± 0.2 d	20.7 ± 0.4 d
Control	0	75.5 ± 0.1 a	-

Numerical values were expressed as mean ± standard error (SE) of triplicates. Different lowercase letters represent significant differences between different biocontrol bacteria at the same concentration (*p* < 0.05).

**Table 5 microorganisms-11-01431-t005:** Antimicrobial activity of natural products.

Natural Products	Concentration (µg/mL)	Regression Equation	EC_50_ (µg/mL)	Coefficient of Determination (R^2^)	95% Confidence Interval
Eugenol	400, 200, 100, 50, 25	y = 0.2825 + 2.4146x	82.39 ± 0.17	0.9904	62.0668–108.4855
Magnolol	400, 200, 100, 50, 25	y = 1.7864 + 1.7121x	75.78 ± 0.22	0.9823	46.5787–121.8670
Thymol	500, 200, 100, 50, 25	y = 2.3047 + 1.3692x	92.84 ± 0.51	0.9217	48.6307–177.8879
Cinnamaldehyde	300, 200, 100, 50, 25	y = 2.0300 + 1.7289x	52.81 ± 0.31	0.9665	36.2457–75.2648
Honokiol	150, 100, 50, 20, 10	y = 1.4328 + 2.1578x	46.18 ± 0.81	0.9539	32.7995–61.7379
Carvacrol	80, 40, 20, 10, 5	y = 1.6919 + 2.4207x	24.19 ± 0.49	0.9514	17.2451–31.3741
2-Allylphenol	60, 30, 20, 10, 5	y = 3.2586 + 2.4329x	7.18 ± 0.02	0.9371	4.3971–11.7511

## Data Availability

The datasets generated and/or analyzed during the study are available from the corresponding author upon reasonable request.

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
