# Peer review of "Identification of a New Pathogenic fungi Causing Sorghum Leaf Spot Disease and Its Management Using Natural Product and Microorganisms"

_microorganisms, 2023, doi:10.3390/microorganisms11061431_

Round 1
Reviewer 1 Report
Comments:
To the Editor Prof.
The manuscript “Identification of a New Pathogenic Fungi Causing Sorghum Leaf Spot Disease and Its Management using Natural Product and Microorganisms, aimed to isolate and identify the causative pathogen of sorghum leaf spot disease. The potential of biocontrol bacteria and natural fungicidal agents to prevent and control sorghum leaf spot disease was also investigated. The study is interesting and of great practical importance. The experimental studies are mostly carried out professionally. The article satisfies the criteria of Microorganisms. As detailed data and interpretation, I recommend it for an international audience in this journal, however some points have to be précised and a minor revision is requested.
1. Abstract, Line 17, mention the accession No. of a new identified strain.
2. Abstract, Line 18, Change, to F. thapsinum using the method of mycelial growth. To against F. thapsinum using the dual culture experiment.
3. Introduction, Line 23, Change, Its seeds…. To Their seeds….
4. Introduction, Line 40, Are the authors isolate these fungi Pestalotiopsis trachycarpicola [1], Alternaria spp [5], Curvularia spp. along with Fusarium spp.
5. Introduction, Line 59, 60, Change, PaeniBacillus TO Paenibacillus.
6. Materials and Methods, Line 129, Change in vitro To italic form.
7. Materials and Methods, Line 133, The control inoculated 0.5 cm filter paper circles with sterile distilled water at four symmetrical points located 20 mm from the pathogen. This not clear in control plates (Fig. 4).
8. Results, Line 244, Change in vitro To italic form.
9. Results, Table 3, I think statistical analyses not true.
10. Results, Line 255, Change Antimicrobial Activity… TO Antifungal Activity….
11. Discussion, Line 292, Rewrite this sentence, Sorghum was widely used, including food, brewing, feed, etc.
12. Discussion, improve the mechanism by which biocontrol bacteria and natural products affect the causal pathogen.
With best regards
Author Response
Dear reviwer
Thank you for your valuable comments on my article. The Manuscript ID is microorganisms-2364137. The title of this manuscript is “Identification of a New Pathogenic Fungi Causing Sorghum Leaf Spot Disease and Its Management using Natural Product and Microorganisms.” I have answered all your comments. In addition, I carefully read the manuscript several times and found and corrected several minor errors.
Point 1: Abstract, Line 17, mention the accession No. of a new identified strain.
Response 1: I have added Strain accession and GenBank Accession.
Line 18
Point 2: “to F. thapsinum using the method of mycelial growth.” > “to against F. thapsinum using the dual culture experiment.”
Response 2: “to F. thapsinum using the method of mycelial growth.” has been changed to “to against F. thapsinum using the dual culture experiment.”
Introduction, Line 23
Point 3: “Its seeds…. “ > “To Their seeds….”
Response 3: “Its seeds…. ” has been changed to “To Their seeds….”
Introduction, Line 40
Point 4: Are the authors isolate these fungi Pestalotiopsis trachycarpicola [1], Alternaria spp [5], Curvularia spp. along with Fusarium spp.
Response 4: No, these are separated from different diseased leaves in different periods.
Introduction, Line 59
Point 5: “PaeniBacillus” > “Paenibacillus”
Response 5: “PaeniBacillus” has been changed to “Paenibacillus”
Materials and Methods, Line 129
Point 6: “in vitro” To italic form
Response 6: “in vitro” has been changed to “in vitro”
Materials and Methods, Line 133
Point 7: The control inoculated 0.5 cm filter paper circles with sterile distilled water at four symmetrical points located 20 mm from the pathogen. This not clear in control plates (Fig. 4).
Response 7: After the filter paper was inoculated, the filter paper was taken down half an hour later. Because the position covered by the filter paper cannot grow normally. If the mycelium cannot grow normally, the picture is not beautiful.
Results, Line 244
Point 8: “in vitro” To italic form
Response 8: “in vitro” has been changed to “in vitro”
Results, Table 3
Point 9: statistical analyses not true
Response 9: I have improved the data.
Results, Line 255
Point 10: “Antimicrobial Activity…” > “Antifungal Activity….”
Response 10: “Antimicrobial Activity…” has been changed to “Antifungal Activity….”
Discussion, Line 292
Point 11: Rewrite this sentence, Sorghum was widely used, including food, brewing, feed, etc.
Response 11: “Sorghum was widely used, including food, brewing, feed, etc.” has rewritten as “Sorghum planting has a long history and a wide range of regions. It is a very important raw material for grain, feed crops, as well as winemaking and vinegarmaking.”
Discussion
Point 12: improve the mechanism by which biocontrol bacteria and natural products affect the causal pathogen
Response 12: I have improved the mechanism by which biocontrol bacteria and natural products affect the causal pathogen.
Reviewer 2 Report
The work entitled ‘Identification of a New Pathogenic Fungi Causing Sorghum 2 Leaf Spot Disease and Its Management using Natural Product 3 and Microorganisms’ was interesting, in fact, the use of biocontrol agents in agriculture is now mandatory and we need more info about new tools to fight the diseases. In general, the concepts are clear and I agree with them, but those are not well exposed and the language must be improved because it is weak in any section of the manuscript. The work needs major revisions, but if the authors could apply the suggestions it may be published.
Abstract
Line 14: it is suggested: the original strain was reisolated after the experimental inoculation.
Introduction
Lines 35-36: These 2 lines must be improved in the language.
Line 36: is induced.
Line 39: to be.
Line 51: found.
Line 58: which could.
Line 61: which could control the…..
Line 62: which could…
Lines 66-71: This period could be in the discussion instead of introduction.
Line 79: antagonistic bacterial strains of .....species....
Lines 81-84: The aim should be more precise, and, let's say, a little bit schematic. Moreover it lacks the activity with the natural products (or essential oils and NP….).
Line 83: spp. instead of app.
M&M
Lines 87-96: A suggestion could be the elimination of all this part and to list, instead of it, the isolates of the pathogen employed in the study. As for antagonistic strains and for essential oils or natural products below. Then it could be added a section describing the isolation and the identification.
Line 103: The isolates should be named and mentioned (see above list of isolates). Moreover, instead of ‘isolated’ to put ‘used for…’. If I understood well. Otherwise it is better to reformulate the sentence.
Line 110: observed with phytopathometric assessments.
Line 130: I would eliminate ‘In short…’ and all similar expression in the text below.
Line 138: …were repeated….
Line 141: At which concentration was the suspension?
Lines 143-150: English language must be improved.
Lines 155-157: The concentrations of solvent and the natural product lack.
In general, concerning the in vitro experiments, which solvent did you use for the natural products? And did you put the solvents at the same concentration as more negative controls?
Results
Line 180: 3.1. Isolation and identification of…etc.
Line 183: Among the 5 isolates….
It lacks a Figure or Table showing the in planta experiment results with the differences of disease severity.
Discussion
In general the English must be improved: one example, significant instead of good results.
Lines 327-334: These are speculations, that was the question in material and ,methods section if you used the negative controls using solvents. In any case all these sentences must be with citations-
Figure and tables captions
In general the captions must be better detailed, the figures should be self explanatory.
In general English must be improved in all sections, as is written in the comments.
Author Response
Response to Reviewer 2 Comments
Dear reviwer
Thank you for your valuable comments on my article. The Manuscript ID is microorganisms-2364137. The title of this manuscript is “Identification of a New Pathogenic Fungi Causing Sorghum Leaf Spot Disease and Its Management using Natural Product and Microorganisms.” I have answered all your comments. In addition, I carefully read the manuscript several times and found and corrected several minor errors. Added some content in the background section.
Line 14
Point 1: it is suggested: the original strain was reisolated after the experimental inoculation.
Response 1: “the original isolate inoculated were reisolated” has been changed to “the original strain was reisolated after the experimental inoculation”
Introduction
Lines 35-36
Point 2:These 2 lines must be improved in the language.
Response 2: I have improved the language of these two lines.
Line 36
Point 3: is induced
Response 3: I've modified it
Line 39
Point 4: to be.
Response 4: I have added “be”.
Line 51
Point 5: found.
Response 5: “fund” has been changed to “found”.
Line 58
Point 6: which could.
Response 6: I have added “which”.
Line 61
Point 7: which could control the…
Response 7: “could control of” has been changed to “which could control the”
Line 62
Point 8: which could…
Response 8: I have added “which”.
Lines 66-71
Point 9: This period could be in the discussion instead of introduction.
Response 9: This period is placed in the introduction to introduce the background of the natural products used in the experiment.
Line 79
Point 10: antagonistic bacterial strains of .....species....
Response 10: I have added species of antagonistic strains.
Lines 81-84
Point 11: The aim should be more precise, and, let's say, a little bit schematic. Moreover it lacks the activity with the natural products (or essential oils and NP….).
Response 11:I have increased the activity of natural products, our aim was to provide a “green” management method against sorghum leaf spot disease, this method is to control diseases through biocontrol bacteria and natural products.
Line 83
Point 12: spp. instead of app.
Response 12: “app.” has been changed to “spp.”
M&M
Lines 87-96
Point 13:A suggestion could be the elimination of all this part and to list, instead of it, the isolates of the pathogen employed in the study. As for antagonistic strains and for essential oils or natural products below. Then it could be added a section describing the isolation and the identification.
Response 13:I have made modifications.
Paenibacillus polymyxa Y-1, Bacillus velezensis MT310,Bacillus amyloliquefaciens MT323, Bacillus subtilis MT332, Bacillus megatherium L2,Bacillus licheniformis MTQ23 all are from the College of Agriculture, Guizhou University. Natural products (eugenol, magnolol, thymol, cinnamaldehyde, honokiol, carvacrol, 2-allylphenol) with purities of ≥98% were purchased from Aladdin Reagent Co., Ltd (Shanghai, China) and stored at 2-8℃.
|
Place |
Time |
Longitude and Latitude |
|
Huaxi City, Guizhou Province, China |
August 2021 |
26°4′N, 106°6′E |
Line 103
Point 14:The isolates should be named and mentioned (see above list of isolates). Moreover, instead of ‘isolated’ to put ‘used for…’. If I understood well. Otherwise it is better to reformulate the sentence.
Response 14:I have reformulate the sentence.
Line 110
Point 15: observed with phytopathometric assessments.
Response 15: I have added “with phytopathometric assessments”.
Line 130
Point 16: I would eliminate ‘In short…’ and all similar expression in the text below.
Response 16: I have removed in short and all similar expression in the text below
Line 138
Point 17: …were repeated….
Response 17: “Were” have been added here
Line 141
Point 18: At which concentration was the suspension?
Response 18: 1 mL of antagonistic bacteria with OD (optical density) = 0.6 was transferred to a 0.25 L flask with 0.1 L nutrient agar broth (NA). “with OD (optical density) = 0.6” have been added.
Lines 143-150
Point 19: English language must be improved.
English language has been improved. “After 48 h of incubation, bacterial solution was filtered using a sterile filter gauze, followed by filtration through a 0.44 µm and a 0.22 µm membrane filter, then, aseptic filtrate was obtained. And the sterile filtrate was added to the medium to concentrations of 25 mL L−1, 50 mL L−1, and 100 mL L−1, respectively, the medium was added to plates, after the medium became cool and solidify, the pathogen discs were inoculated to the center of PDA plates containing. After incubation under 25℃ for 7 days, the colony diameter was measured using the "ten" crossing method. All experiments repeated three times. Adding sterile water was used as a control. The inhibitory rate was calculated according: inhibitory rate (%) = [(dcontrol − dtreatment)/dcontrol] × 100 [31], d is the diameter of the F. thapsinum colony”.
Lines 155-157
Point 20: The concentrations of solvent and the natural product lack.
Response 20: The concentrations of the natural product is in Table 4
The concentrations of solvent have been added “all solvents have a concentration greater than 99%”
We set up corresponding controls, and the solvent had no significant effect on colony growth.
Results
Line 180
Point 21: 3.1. Isolation and identification of…etc.
Response 21: I have made the corresponding changes.
Line 183:
Point 22: Among the 5 isolates….
It lacks a Figure or Table showing the in planta experiment results with the differences of disease severity.
Response 22: “Among the 5 isolates” has been added.
Discussion
Point 23:
In general the English must be improved: one example, significant instead of good results.
Lines 327-334: These are speculations, that was the question in material and, methods section if you used the negative controls using solvents. In any case all these sentences must be with citations-
Figure and tables captions
In general the captions must be better detailed, the figures should be self explanatory.
Response 23: English has been improved. All controls were added with the corresponding solvent. Figure and tables captions has been with cited